# A universal deep neural network for in-depth cleaning of single-cell RNA-Seq data

Hui Li[1,2], Cory R. Brouwer [1,2] & Weijun Luo [1,2,3✉]

Single cell RNA sequencing (scRNA-Seq) is being widely used in biomedical research and generated enormous volume and diversity of data. The raw data contain multiple types of noise and technical artifacts, which need thorough cleaning. Existing denoising and imputation methods largely focus on a single type of noise (i.e., dropouts) and have strong distribution assumptions which greatly limit their performance and application. Here we design and develop the AutoClass model, integrating two deep neural network components, an autoencoder, and a classifier, as to maximize both noise removal and signal retention. AutoClass is distribution agnostic as it makes no assumption on specific data distributions, hence can effectively clean a wide range of noise and artifacts. AutoClass outperforms the state-of-art methods in multiple types of scRNA-Seq data analyses, including data recovery, differential expression analysis, clustering analysis, and batch effect removal. Importantly, AutoClass is robust on key hyperparameter settings including bottleneck layer size, pre-clustering number and classifier weight. We have made AutoClass open source at: https://github.com/datapplab/AutoClass.

[1] Department of Bioinformatics and Genomics, College of Computing and Informatics, UNC Charlotte, Charlotte, NC 28223, USA. [2] UNC Charlotte Bioinformatics Service Division, North Carolina Research Campus, Kannapolis, NC 28081, USA. [3] Present address: Novant Health, Charlotte, NC 28207, USA. ✉email: luo_weijun@yahoo.com

ScRNA-Seq has been widely adopted in biological and medical research[1–5] as an ultra-high resolution and ultra-high throughput transcriptome profiling technology. Enormous amount of data has been generated providing great opportunities and challenges in data analytics.

First of all, scRNA-Seq data come with multiple types of noise and quality issues. Some are issues associated with gene expression profiling in general, including RNA amplification bias, uneven library size, sequencing and mapping error, etc. Others are specific to single cell assays. For example, extremely small sample quantity and low RNA capture rate result in large number of false zero expression or dropout[6]. Individual cells vary in differentiation or cell cycle stages[7], health conditions, or stochastic transcription activities, which are biological differences but irrelevant in most studies. In addition, substantial batch effects are frequently observed[8] due to inconsistence in sample batches and experiments. Most of these noises and variances are not dropout and may follow Gaussian, Poisson, or more complex distributions depending on the source of the variances. All of these variances need to be corrected and cleaned so that biologically relevant differences can be reconstructed and analyzed accurately.

Multiple statistical methods have been developed to impute and denoise scRNA-Seq data. Most of these methods rely on distribution assumptions on scRNA-Seq data matrix. For example, deep count autoencoder (DCA)[9] assumes negative binomial distribution with or without zero inflation, SAVER[10] assumes negative binomial distribution, and scImpute[11] uses a mixture of Gaussian and Gamma model. Currently, there is no consensus on the distribution of scRNA-Seq data. Method with inaccurate distribution assumptions[12] may not denoise properly, but rather introduce new complexities and artifacts. Importantly, these methods largely focus on dropouts and ignore other types of noise and variances, which hinders accurate analysis and interpretation of the data.

To address these issues, we developed AutoClass, a neural network-based method. AutoClass integrates two neural network components: an autoencoder and a classifier (Fig. 1a and "Methods"). The autoencoder itself consists of two parts: an encoder and a decoder. The encoder reduces data dimension and compresses the input data by decreasing hidden layer size (number of neurons). The decoder, in the opposite, expands data dimension and reconstructs the original input data from the compressed data by increasing hidden layer size. Note the encoder and decoder are symmetric in both architecture and function. The data is most compressed at the so-called bottleneck layer between the encoder and the decoder. The autoencoder itself, as an unsupervised data reduction method, is not sufficient in separating signal from noise (Fig. 1b). To ensure the encoding process filter out noise and retain signal, we add a classifier branch from the bottleneck layer (Fig. 1a and "Methods"). Instead of known cell classes, virtual class labels are generated by pre-clustering. Therefore, AutoClass is a composite deep neural network with both unsupervised (autoencoder) and supervised (classifier) learning components. Like regular autoencoder methods, AutoClass is unsupervised or self-supervised because true data or labels are not used in training. AutoClass does not presume any specific type or form of data distribution, hence has the potential to correct a wide range noises and non-signal variances. In addition, it can model non-linear relationships between genes with non-linear activation functions. In this study, we extensively evaluated AutoClass against existing methods using multiple simulated and real datasets. We demonstrated AutoClass can better reconstruct scRNA-Seq data and enhance downstream analysis in multiple aspects. In addition, AutoClass is robust over hyperparameter settings and the default setting applies well in various datasets and conditions.

## Results

**Validation of the classifier component.** The unique part of AutoClass is the classifier branch from the bottleneck layer. Since encoding process losses information in the input data, the classifier branch is added to make sure relevant information or signal is sufficiently retained. To show that the classifier is needed, we simulated a scRNA-Seq Dataset 1 (see "Methods" and Supplementary Table 2) using Splatter[13] with 1000 genes and 500 cells in six groups, with and without dropout. Applied both AutoClass and a regular autoencoder without the classifier on the data with dropout, the results are illustrated in two-dimensional t-SNE (see "Methods") plots in Fig. 1b. AutoClass but not the regular autoencoder was able to recover cell type pattern, indicating the classifier component is necessary for reconstructing scRNA-Seq data.

**Gene expression data recovery.** We evaluated expression value recovery on simulated scRNA-Seq data with different noise types or distributions. We generated and scRNA-Seq dataset using Splatter with 500 cells, 1000 genes in five cell groups with (raw data, Dataset 2) and without dropout (true data). From the same true data, we also generated 5 additional raw datasets by adding noise following different distributions which are representative and commonly seen, including random uniform (Dataset 3), Gaussian (Dataset 4), Gamma (Dataset 5), Poisson (Dataset 6) and negative binomial (Dataset 7) (details in "Methods" and Supplement Tables 2 and 3).

As expected, dropout noise greatly reduced the data quality and obscured the signal or biological differences such as distinction between cell types (Fig. 2a). All other noise types had similar effect on the data (Fig. 2b, c and Supplementary Fig. 1). With t-SNE transformation on Dataset 2–7, the true data without noise showed distinct cell types, but not the raw data with noises (Fig. 2a–c and Supplementary Fig. 1). The average Silhouette width[14] (ASW) on the t-SNE plot is a measurement of distance between groups, ranges from −1 to 1, where higher values indicate more confident clustering. ASW dropped greatly from 0.64 to around 0 in all raw datasets. After imputation by AutoClass, the cell type pattern was recovered and ASW increased back substantially to 0.2–0.5. In contrast, all published control methods (DCA, MAGIC[15], scImpute, and SAVER) were unable to recover the original cell type pattern (Fig. 2a–c and Supplementary Fig. 1) and ASW scores remained low (Fig. 2d) for all noise types.

We also measure the data recovery quality using other metrics. The mean squared error (MSE) between the true data and imputed/denoised data for Dataset 3–7 (5 noise types other than dropout) were also computed (Fig. 2e). Among the five tested methods, AutoClass consistently achieved the smallest MSE for all noise types (Fig. 2e). Dropout noise (Dataset 2) is very different from all other noise types (Dataset 3–7) in both distribution form and generation mechanism, and MSE was not an informative measurement of data recovery. We computed the average recovered values of dropout zeros and those of true zeros (Fig. 2f) instead. An ideal imputation method can distinguish between these two types of zeros, i.e., impute dropout zeros while retain true zeros (Fig. 2f). While SAVER was too conservative in imputing both types of 0 values, DCA and MAGIC were too aggressive. AutoClass and scImpute both achieved good balance between imputing dropout 0 s and retaining true 0 s, yet only the former but both the later was able to recover the biological difference or distinct cell type clustering (Fig. 2a–c and Supplementary Fig. 1).

**Differential expression analysis.** Differential expression (DE) analysis is by far the most common analysis of scRNA-Seq and gene expression data. To study the performance of AutoClass in DE analysis, we simulated a scRNA-Seq Dataset 8 using Splatter

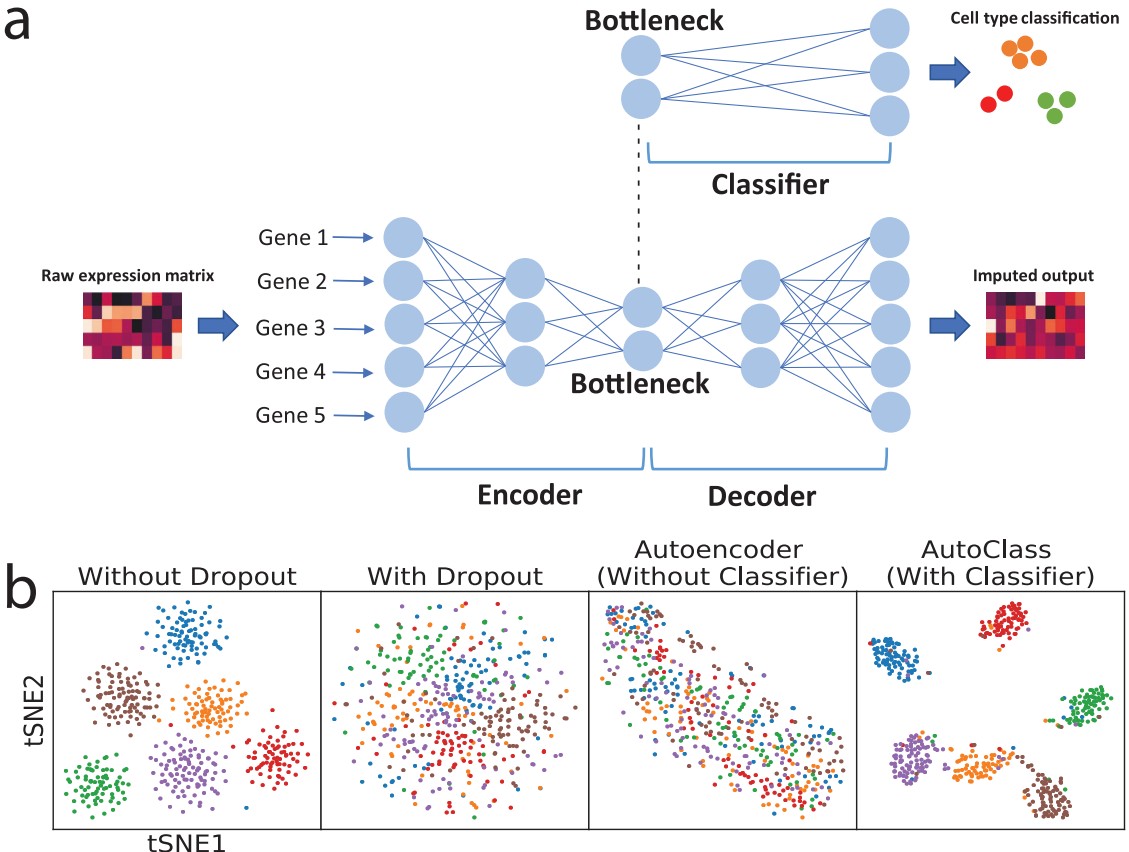

**Fig. 1 AutoClass integrates a classifier to a regular autoencoder, as to fully reconstruct scRNA-Seq data. a** AutoClass consists a regular autoencoder and a classifier branch from the bottleneck layer. The input raw expression data is compressed in the encoder, and reconstructed in the decoder, the classifier branch helps to retain signal in data compression. The output of the autoencoder is the desired imputed data (see "Methods" for details). **b** t-SNE plots of Dataset 1 without dropout, with dropout, with dropout imputed by a regular autoencoder and AutoClass.

with 1000 genes and 500 cells in two cell groups. Here the ground truth of 161 truly differentially expressed genes is known. We applied Two-sample T-test to the true, raw, and imputed data using different methods. The median value of t-statistics for the truly differentially expressed genes dropped from 5.79 in the true data to 2.11 in the raw data, and increased back to 5.86 upon imputation by AutoClass, which was almost the same as in the true data and higher than in all control methods (Fig. 3a, b). As shown by ROC curves and area under the curves (AUC), Auto-Class also was the best at balancing true positives and false negatives (Fig. 3c, d). At specificity = 0.90 or 1-specificity = 0.10 (dashed vertical line in Fig. 3c), the ROC curves marked different levels of sensitivity, i.e., 0.72 (True data), 0.61 (AutoClass), 0.52 (DCA), 0.51 (scImpute), 0.41 (MAGIC), 0.35 (SAVER) and 0.30 (Raw data). AutoClass was the best method in this analysis, and achieved the closest performance to the True data.

Similarly, AutoClass can improve DE analysis in data with Gaussian noise. We manually added Gaussian noise to the true data of Dataset 8 to generate the raw data of Dataset 9. The DE analysis results can be found in Supplementary Fig. 2.

AutoClass also improves marker gene expression analysis. Baron dataset[16] provides known marker gene lists for related cell types in pancreatic islets. AutoClass is the only method consistently improved DE analysis results for the marker genes in both fold changes and t-statistics. The marker genes reached the highest median fold change ($2^{3.333} = 10.1$) after imputation by AutoClass (vs 7.9 in the raw data and 7.5–9.2 by other methods, Fig. 3e). Note that MAGIC has slightly higher median t-statistics (Fig. 3f) than AutoClass, but the difference is not significant statistically ($p = 0.3$).

AutoClass imputation helps to identify more potential marker genes. For example, three differentially expressed genes in Supplementary Fig. 3 were selected in AutoClass imputed data, but not in raw data or imputation by other methods, including RGS2 in delta cells, SLC7A2 in alpha cells[17], and S100A10 in ductal cells[18]. After imputation by AutoClass, the expression patterns became more distinct (column 3 vs 2 in Supplementary Fig. 3) and expression curves in the target cell type vs other cell types became better separated (green curves vs orange curves in columns 4–5, Supplementary Fig. 3).

**Clustering analysis**. Clustering analysis is frequently done on scRNA-seq data as to identify cell types or subpopulations. To evaluate AutoClass for clustering analysis, we used four real datasets, including two small datasets: the Buettner dataset[2] (182 cells) and the Usoskin dataset[19] (622 cells) and two large datasets: the Lake dataset[20] (8592 cells) and the Zeisel dataset[21] (3005 cells). Detailed information for these datasets can be found in Methods and Supplementary Table 1.

We compared K-means clustering results on the 200 highest variable genes. The ground truth or the actual number of cell types were used as a number of clusters. Clustering results were evaluated by four different metrics: adjusted Rand index[22] (ARI), Jaccard Index[23] (JI), normalized mutual information[24] (NMI), and purity score[25] (PS). All of them range from 0 to 1, with 1 indicating a perfect match to the true groups. AutoClass is the only method improving all four metrics from the raw data. In addition, AutoClass achieved the best clustering results for 3 out of 4 datasets (Table 1).

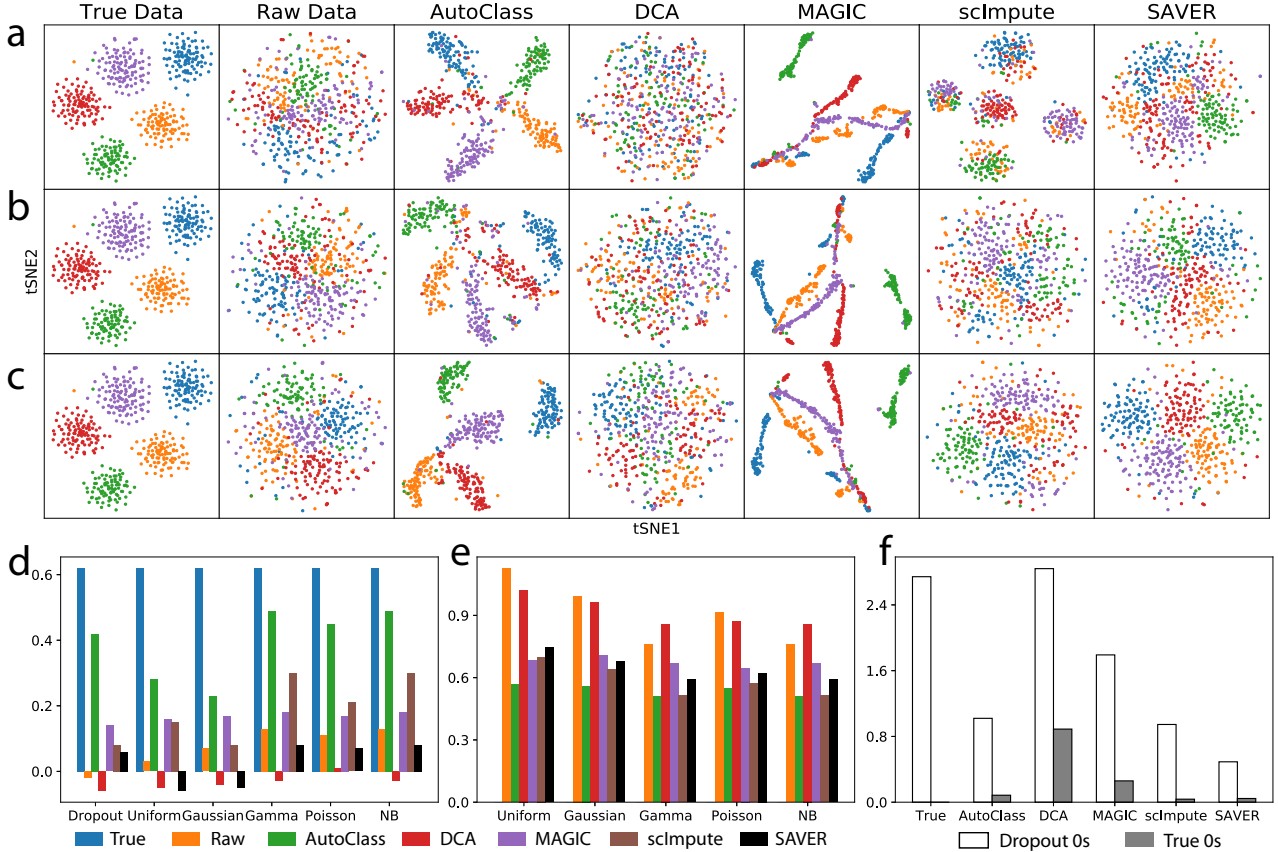

**Fig. 2 Gene expression data recovery after imputation. a–c** t-SNE plots for Dataset 2 (dropout noise), Dataset 4 (Gaussian noise) and Dataset 7 (negative binomial noise), respectively. **d** Average Silhouette width based on t-SNE plot for Dataset 2-7. **e** Mean squared error between true data and imputed data for Dataset 3-7. **f** Average recovered values of dropout 0 s and true 0 s for different imputation methods.

For the Usoskin dataset, out of all tested methods, only AutoClass and MAGIC reconstructed distinct clusters (Fig. 4a). But MAGIC likely generated false positive signals, given that the between-group cell-to-cell correlation are almost the same as within-group correlation, and both are close to 1 (Fig. 4b). AutoClass was the only method differentiating within-group vs between-group correlation as informative metrics for signal vs noise (Fig. 4b).

**Batch effect removal**. Batch effect rises from different individual cell donors, sample groups, or experiment conditions and can severely affect downstream analysis. We analyzed two real datasets with major batch effect. The Villani dataset[26] sequenced 768 human blood dendritic cells (DC) in 2 batched using Smart-Seq2. The Baron dataset includes 7162 pancreatic islet cells from three healthy individuals.

Similar to Tran et al.[8], we evaluated the performance of batch effect correction as the ability to merge different batches of the same cell type while keeping different cell types separate. We did t-SNE transformation on the data first (Fig. 5a and Supplementary Fig. 4), then applied the four metrics above mentioned, i.e., ASW, ARI, NMI, and PS on both cell types and batches. While cell-type-level metrics measure cell type separation, 1 - batch level metrics measure the merging between batches of same cell type (Fig. 5b and Supplementary Fig. 5).

In the Villani dataset (Fig. 5), the raw data shows clear separation in both cell types and sample batches. After imputation by AutoClass, while cell types remained well separated, the two batches were evenly mixed up within each cell type. In contrast, SAVER

failed to reduce the batch effect, while all other methods even aggravated it (Fig. 5).

Note that AutoClass corrects the batch effect without knowing the actual number of cell types. Here, we used the default number of clusters in the pre-clustering step, i.e., [8, 9, 10] (see "Methods"). This is close to the number of spurious groups counting batches (i.e., 8), but far away from the actual number of cell types, or 4. In other words, AutoClass was not misled by the pre-clustering number and correctly recovered the actual cluster number.

In Baron dataset (Supplementary Figs. 4, 5), AutoClass reduced the batch effect and increased cell type separation simultaneously with the default pre-clustering number too. MAGIC dramatically reduced the differences in both batches and cell type. The batch effect correction by other methods were limited.

**Robustness over major hyperparameters**. AutoClass, as a composite deep neural network, has multiple hyperparameters. Among them, the most important ones are bottleneck layer size, number of pre-clusters and classifier weight. Bottleneck layer plays an important role in autoencoders, it is the narrowest part of the network and the size (number of neurons) controls how much the input data is compressed. The number of clusters ($K$) in the pre-clustering step is specific to the classifier of AutoClass. AutoClass uses three consecutive cluster numbers $[K-1, K, K+1]$, and the final imputation output is the average over three predictions using these three clustering numbers (see "Methods"). In addition, the classifier weight $w$ (see Eq. 4 and "Methods") is another AutoClass specific hyperparameter which balance the ratio between autoencoder loss and the classifier loss.

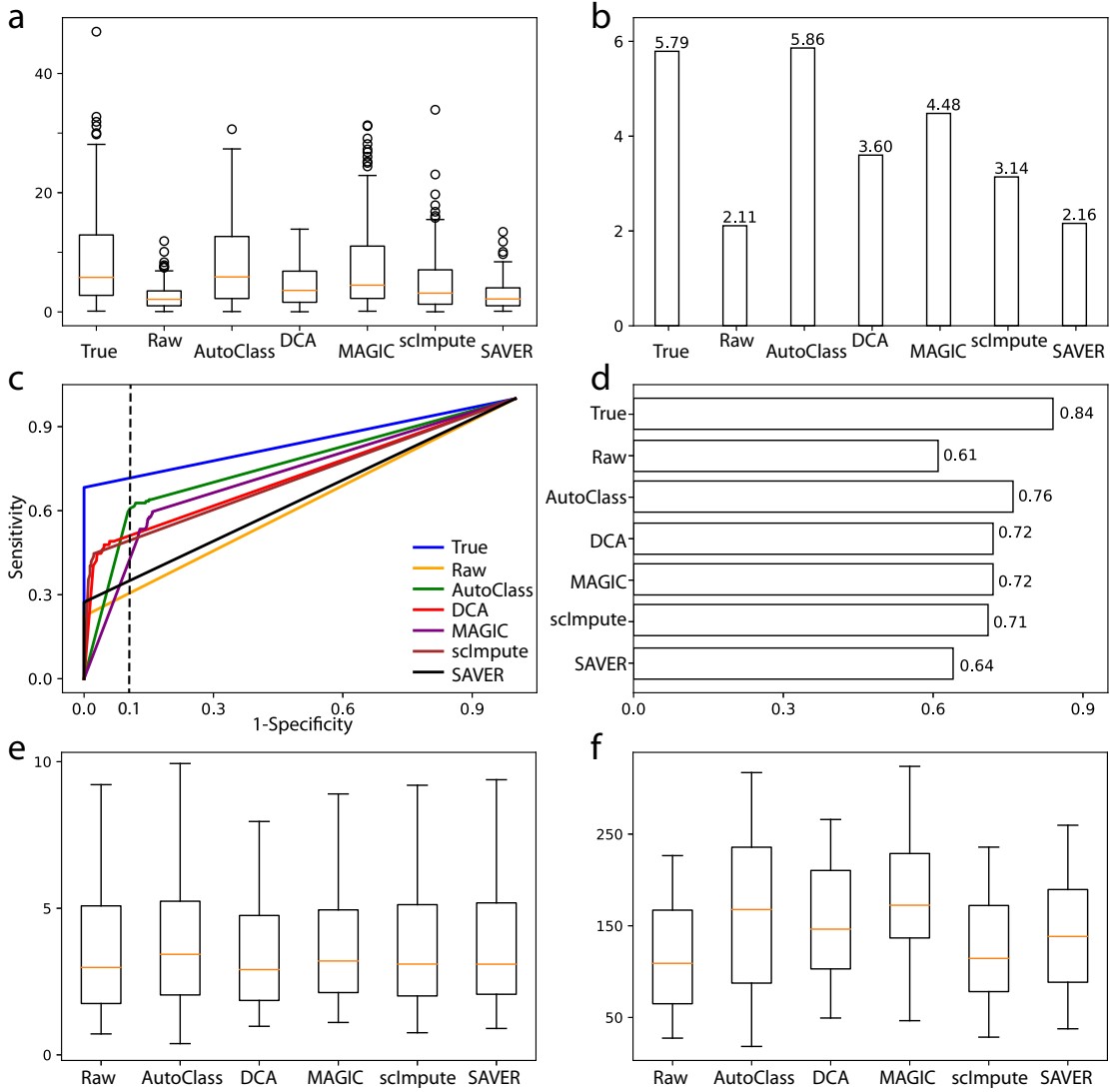

**Fig. 3 Differential expression analysis and marker gene analysis. a**, **b** T-statistics and their median values for truly differentially expressed genes in Dataset 8. **c**, **d** ROC curves and areas under the ROC curves for Dataset 8. **e**, **f** log2 based fold changes and t-statistics of marker genes in the Baron dataset. In **a**, **e**, and **f**, the box represents the interquartile range, the horizontal line in the box is the median, and the whiskers represent 1.5 times the interquartile range, with sample size $n = 53$ marker genes.

AutoClass is robust over a wide range of bottleneck layer sizes, pre-clustering $K$ values (Fig. 6 and Supplementary Figs. 6, 7) and classifier weight $w$ (Supplementary Fig. 8). The t-SNE clustering patterns, clustering metrics (ASW and ARI), MSE and imputed dropout0s/true 0 s ratio remained the same when bottleneck layer size increase from 16 to 256 (Fig. 6a, c and Supplementary Fig. 7a). However, these results or metrics varied heavily in the same analysis using DCA, another autoencoder based method (Fig. 6b, c and Supplementary Fig. 7a). Likewise, AutoClass also achieved stable results over the range of $K$ values - 4–8 (Fig. 6d and Supplementary Figs. 6, 7b) and the range of classifier weight $w$ values - 0.1–0.9 (Supplementary Fig. 8).

Interestingly, AutoClass is robust on the choice of loss function for the autoencoder part or reconstruction error. In Eq. 5, instead of using $\left| \overline{\mathbf{X}} - \mathbf{Y}_k \right|^2$ or $\mathrm{MSE}(\overline{\mathbf{X}}, \mathbf{Y}_k)$, we can also use $\left| \overline{\mathbf{X}} - \mathbf{Y}_k \right|^p$ ($p = 1, 2, 3, ..$). Supplementary Figure 10 shows the t-SNE plots for AutoClass denoising results of simulated Dataset 2 (dropout noise), Dataset 3 (uniform noise) and Dataset 7 (negative binomial noise) for different values of the $p$ parameter Indeed, AutoClass successfully recovered cell types or biological signals

from different types or distributions of noise using different values of $p$ or reconstruction errors.

Note the loss function becomes a mean absolute error (MAE) when $p = 1$. While MSE or $p = 2$ coincides with normal distribution assumption, all other $p$ parameter values had no implication of specific distribution assumption. Note our reconstruction error in Eq. 5 measures the model misfit in general and is not log likelihood. Both the reconstruction error in Eq. 5 and the overall loss in Eq. 4 are data driven, and have no assumption of prior knowledge on the forms of the underlying data distribution (distribution agnostic). Therefore, AutoClass is distribution agnostic and works independent of both the $p$ parameter value here and the noise types or distributions (Fig. 2a–c and Supplementary Fig. 1). We set $p = 2$ (MSE loss) as default since this is most commonly used for reconstruction error.

**Scalability**. To evaluate the scalability of AutoClass vs control methods, we simulated a series of scRNA-Seq data (Scalability datasets) with six sample sizes, i.e., 1000, 2000, 4000, 8000, 16,000 and 32,000 cells, and with 1000 genes each in Splatter. Then the

**Table 1 Evaluation of clustering results of four real scRNA-Seq datasets.**

| Metric | Dataset | Raw | AutoClass | DCA | MAGIC | scImpute | SAVER |
|---|---|---|---|---|---|---|---|
| ARI | Buettner | 0.023 | **0.372** | 0.288 | 0.213 | 0.039 | 0.016 |
| | Usoskin | 0.221 | **0.869** | 0.234 | 0.813 | 0.067 | 0.317 |
| | Lake | 0.403 | 0.557 | **0.572** | 0.440 | 0.313 | 0.465 |
| | Zeisel | 0.737 | **0.793** | 0.753 | 0.433 | 0.623 | 0.763 |
| JI | Buettner | 0.242 | **0.409** | 0.363 | 0.368 | 0.262 | 0.247 |
| | Usoskin | 0.324 | **0.830** | 0.284 | 0.764 | 0.266 | 0.351 |
| | Lake | 0.323 | 0.439 | **0.453** | 0.346 | 0.254 | 0.364 |
| | Zeisel | 0.646 | **0.713** | 0.664 | 0.370 | 0.679 | 0.677 |
| NMI | Buettner | 0.035 | **0.395** | 0.333 | 0.335 | 0.075 | 0.038 |
| | Usoskin | 0.225 | **0.829** | 0.253 | 0.771 | 0.048 | 0.431 |
| | Lake | 0.611 | 0.667 | **0.676** | 0.601 | 0.500 | 0.642 |
| | Zeisel | 0.747 | **0.784** | 0.746 | 0.598 | 0.798 | 0.762 |
| PS | Buettner | 0.434 | **0.720** | 0.648 | 0.599 | 0.445 | 0.423 |
| | Usoskin | 0.545 | **0.937** | 0.579 | 0.913 | 0.416 | 0.682 |
| | Lake | 0.723 | **0.772** | 0.766 | 0.693 | 0.610 | 0.742 |
| | Zeisel | 0.894 | **0.917** | 0.880 | 0.763 | 0.548 | 0.897 |

The four metrics are adjusted Rand index (ARI), Jaccard Index (JI), normalized mutual information (NMI), and purity score (PS). Highest value in each row was highlighted in boldface.

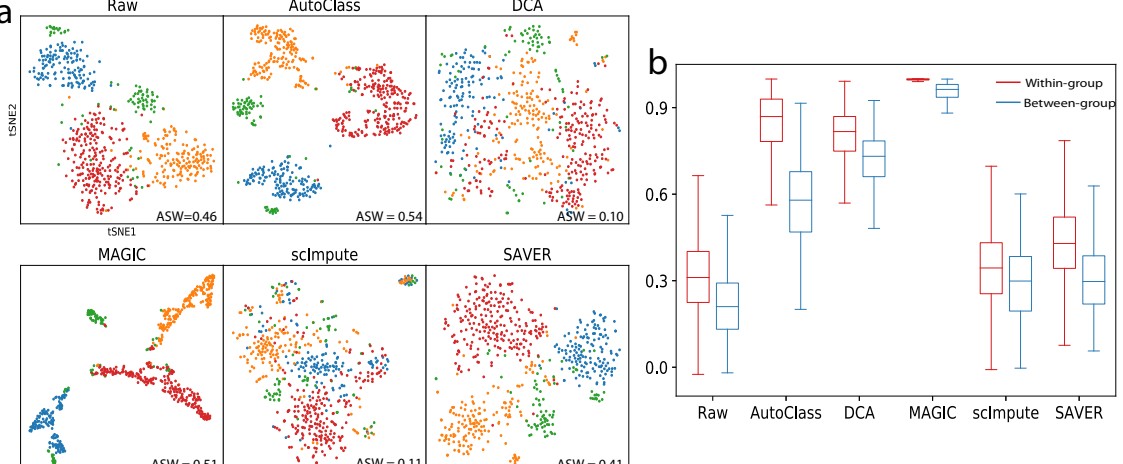

**Fig. 4 Imputation results for the Usoskin dataset. a** t-SNE plots for raw and imputed data. **b** Within-group and between-group cell-to-cell correlation for raw and imputed data. In **b**, the box represents the interquartile range, the horizontal line in the box is the median, and the whiskers represent 1.5 times the interquartile range, with sample size $n = 50,000$ randomly sampled cell pairs.

data were denoised using AutoClass and control methods with their runtime recorded (Fig. 7).

AutoClass was highly efficient and scalable. It runs fast on a regular laptop (8-core Intel Core i5-8265U CPU at 1.60 GHz, 8 G RAM), and processed 1000 cells in 20 s, 8000 cells in 119 s, and 32,000 cells in 706 s (Fig. 7). The runtime scaled almost linearly with the number of cells. AutoClass is consistently faster than DCA, another deep neuron network based method, and 2–3 orders of magnitude faster than SAVER and scImpute. Only MAGIC runs faster than AutoClass, but the difference shrinks quickly when cell number increases. Our observations on runtime were largely consistent with the evaluation in DCA paper[9], and the minor differences likely reflect the different hardware settings or software versions.

**Large feature size and sample size.** Previous analyses were performed on 1000 highly variable. It is proper to focus on a selected subset of all genes in scRNA-Seq, because: (1) many genes that are mostly zeros or hardly changes across cells are non-informative and contribute little information to downstream analyses or denoising, and it would be a waste of time/resource to

impute these genes; (2) we are able to complete benchmark experiments on all methods (including the slow ones shown in Fig. 7) in limited computing time and resource. Therefore, Many existing methods focus on subset of genes explicitly[9,10].

However, AutoClass applies equally well to datasets with larger number of genes when necessary. As an example suggested by a reviewer, we worked on a benchmarking dataset from Tian et al.[27], which sequenced a mixture of five cell lines (with 305 cells and 13,426 genes). We chose to work with either the 1000 or 5000 most variable genes (Fig. 8). Note the original raw data has high quality and very distinct cell clusters without any imputation. To facilitate the performance evaluation of different methods, negative binomial noise or dropouts were added to the dataset (Supplementary Tables 2–3). AutoClass is the only method that consistently recovers the original cell clusters or the biological differences from the noisy data, and it also achieved the highest ASW scores across different noise types and gene numbers (Fig. 8). Note that the performance of AutoClass was similar or better when more genes were included or feature size became larger (5000 vs 1000 genes).

AutoClass applies well to datasets with both larger features size (genes) and sample size (cells). Because SAVER and scImpute

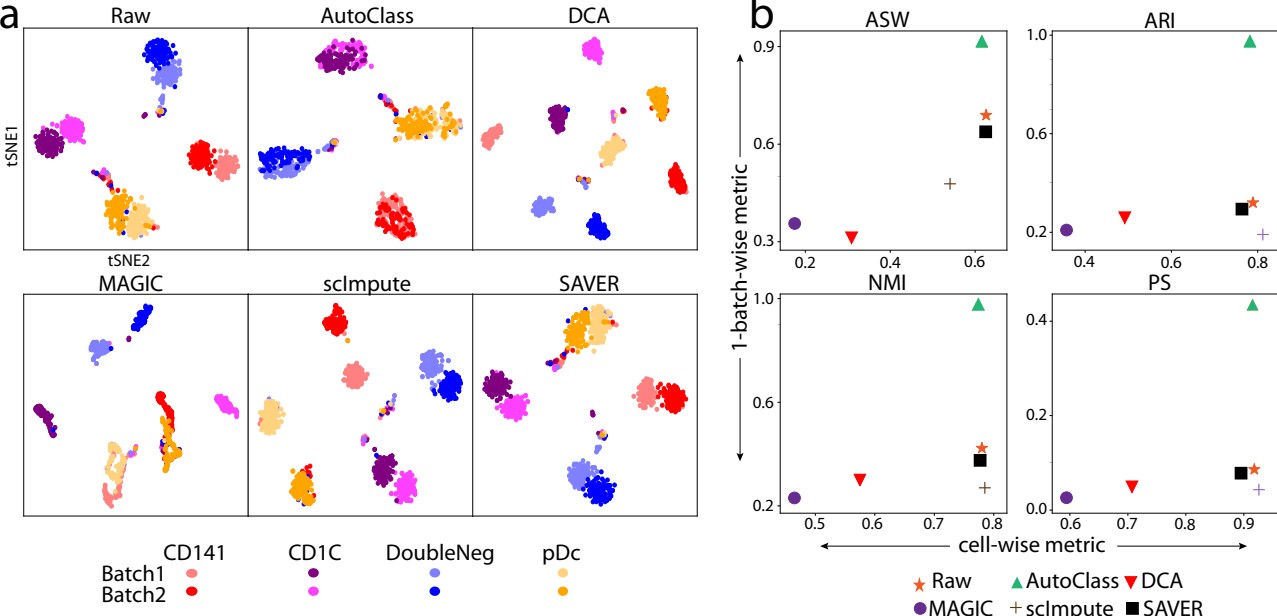

**Fig. 5 Batch effect removal in the Villani dataset. a** t-SNE plots for raw and imputed data. **b** Evaluation of batch and cell type separation in raw and imputed data by four different metrics: average Silhouette width, adjusted Rand index, normalized mutual information, and purity score. Good performance is shown as big values in both X-(cell type separation) and Y-(batch effect removal) axes.

were much slower (Fig. 7), and could not complete these tasks in a reasonable amount of time on our working machine, we only include AutoClass, DCA and MAGIC in these analyses. These two simulated datasets have 10,000 genes and 10,000 cells in 7 groups each, generated using either Splatter[13] or an alternative method used by scImpute[11]. Again, AutoClass was the only method that consistently improves the data quality and recovers the original cell clusters from the noisy raw data (Supplementary Fig. 11). In contrast, the performance of both DCA and MAGIC failed to improve the data quality in Dataset 10 or even made it worse.

All these experiments consistently show that AutoClass works with different sample sizes, feature sizes, in addition to different noise types and data distributions.

## Discussion

In this work, we proposed and developed a deep learning-based method AutoClass for thorough cleaning of scRNA-Seq data. AutoClass integrates two neural network components, an autoencoder, and a classifier. This composite network architecture is essential for filtering out noise and retaining signal effectively. Unlike many other scRNA-Seq imputation methods, AutoClass does not rely on any distribution assumption, and fully counts the non-linear interactions between genes. With these properties, AutoClass effectively models and cleans a wide range of noises and artifacts in scRNA-Seq data including dropouts, random uniform, Gaussian, Gamma, Poisson, and negative binomial noises, as well as batch effects. These are the most common and representative types of noises and artifacts. Any other types not directly tested would likely be cleaned with the same efficiency because they are similar in distribution and source and AutoClass has no assumption on the noise forms. Such in-depth cleaning led to consistent and substantial improvement of the data quality and downstream analyses including differential expression and clustering, as shown by a range of experiments with both simulated and real datasets.

Note that even though we use MSE as the default loss function, AutoClass takes no assumption on the noise type or distribution. Indeed, it works equally well with other loss functions in the general form of $\left| \overline{\mathbf{X}} - \mathbf{Y}_k \right|^p$ ($p = 1, 2, 3$, etc, Supplementary Fig. 10), and a wide range of noise distributions other than Guassian.

Hyperparameter tuning is an important yet tedious step for training neural network models. Inadequate tuning of hyperparameters may lead to suboptimal results. Remarkably, AutoClass is robust with key hyperparameters including bottleneck layer size ($n$), pre-clustering number ($K$), and classifier weight ($w$). The default setting with $n = 128, K = 9, w = 0.9$ works well for most scRNA-Seq datasets and conditions. This robustness makes AutoClass an appealing method for both performance and practical uses.

AutoClass is highly efficient and scalable. It easily fits a personal laptop and processes thousands of scRNA-Seq samples in a few minutes or even less time. As shown in the series of experiments using various real and simulated datasets in this study (Supplementary Tables 1–3), AutoClass works well with data of a wide range of sample sizes, feature sizes, or both consistently.

## Methods

**Architecture of AutoClass**. AutoClass integrates two neural network components, an autoencoder, and a classifier, to impute scRNA-seq data (Fig. 1a). The classifier branch is necessary to preserve signals or biological differences (cell type patterns etc.) from loss in data compression by the encoder.

When cell classes are unknown, virtual class labels are generated by pre-clustering using K-means method. The total loss of the entire network is the weighted sum of classifier loss (cross-entropy or CE) and the autoencoder loss or reconstruction error (RE). The activation functions in the hidden layers are all rectified linear unit (ReLU), the activation functions for the output layer of autoencoder and classifier are SoftPlus and SoftMax, respectively.

The formulation of AutoClass architecture is:

$$\mathbf{B}_k = \mathrm{Encoder}(\overline{\mathbf{X}}) \qquad (1)$$

$$\mathbf{Y}_k = \mathrm{Decoder}(\mathbf{B}_k) \qquad (2)$$

$$\mathbf{C}_k = \mathrm{Classifier}(\mathbf{B}_k) \qquad (3)$$

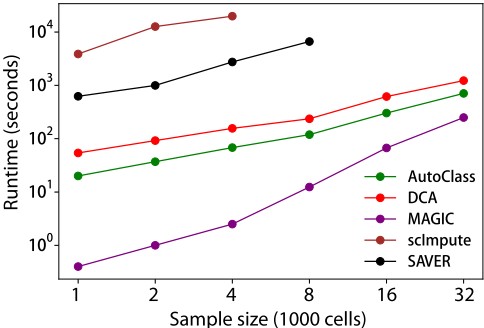

**Fig. 6 Impact of bottleneck layer size and the number of clusters in the pre-clustering in Dataset 1.** t-SNE plots of data imputed by AutoClass (**a**) and DCA (**b**) with different bottleneck sizes (*n*). Imputation results by AutoClass with bottleneck layer size (**c**) and the number of clusters in the pre-clustering (**d**).

**Fig. 7** Sample size vs runtime for different denoising methods.

$$L_k = w \times \text{CE}(\widehat{\mathbf{C}}_k, \mathbf{C}_k) + (1-w) \times \text{RE}(\overline{\mathbf{X}}, \mathbf{Y}_k) \quad (4)$$

$$\text{RE} = |\overline{\mathbf{X}} - \mathbf{Y}_k|^p \quad (5)$$

Where $\mathbf{B}_k, \mathbf{Y}_k, \mathbf{C}_k$ and $L_k$ are the bottleneck representation, the output of the decoder hence the autoencoder, the output of the classifier, and total loss, respectively. $\widehat{\mathbf{C}}_k$ is the pre-clustering cell type labels for $k$ clusters. $\overline{\mathbf{X}}$ is the input of AutoClass, and has been normalized over library size and followed by a $\log_2$

transformation with pseudo count 1:

$$\overline{\mathbf{X}} = \log_2\left(\text{diag}(\mathbf{s}_i)^{-1}\mathbf{X} + 1\right) \quad (6)$$

$\mathbf{X}$ is the raw count matrix and the size factor $\mathbf{s}_i$ for cell $i$ is equal to the library size divided by the median library size across cells. Library size is defined as the total number of counts per cell.

The final imputed data is the average prediction of the autoencoder over different cluster numbers in pre-clustering:

$$\mathbf{Y} = \text{E}\left(\mathbf{Y}_k | k\right) \quad (7)$$

For all datasets in this manuscript, we used 3 consecutive cluster numbers, or $k = [K-1, K, K+1]$, the default value is $K = 9$. Although AutoClass has the option to use existing cell type labels instead of pre-clustering when proper, pre-clustering is the default and was used in all AutoClass analyses in this work. All known cell type labels were used for method evaluation only. The final imputation result was the average results over different $K$s.

**AutoClass Implementation and hyperparameter settings.** AutoClass is implemented in Python 3 with Keras. *Adam* is used for optimizer with default learning rate 0.001. Learning rate is multiplied by 0.1 if validation loss does not improve for 15 epochs. The training stops if there is no improvement for 30 epochs.

Although AutoClass works well for small bottleneck layer sizes ($n = 16, 32$ or similar), we set the default value to be $n = 128$, as be conservative and to avoid potential information loss in data compression. This default value was used in all datasets in this study.

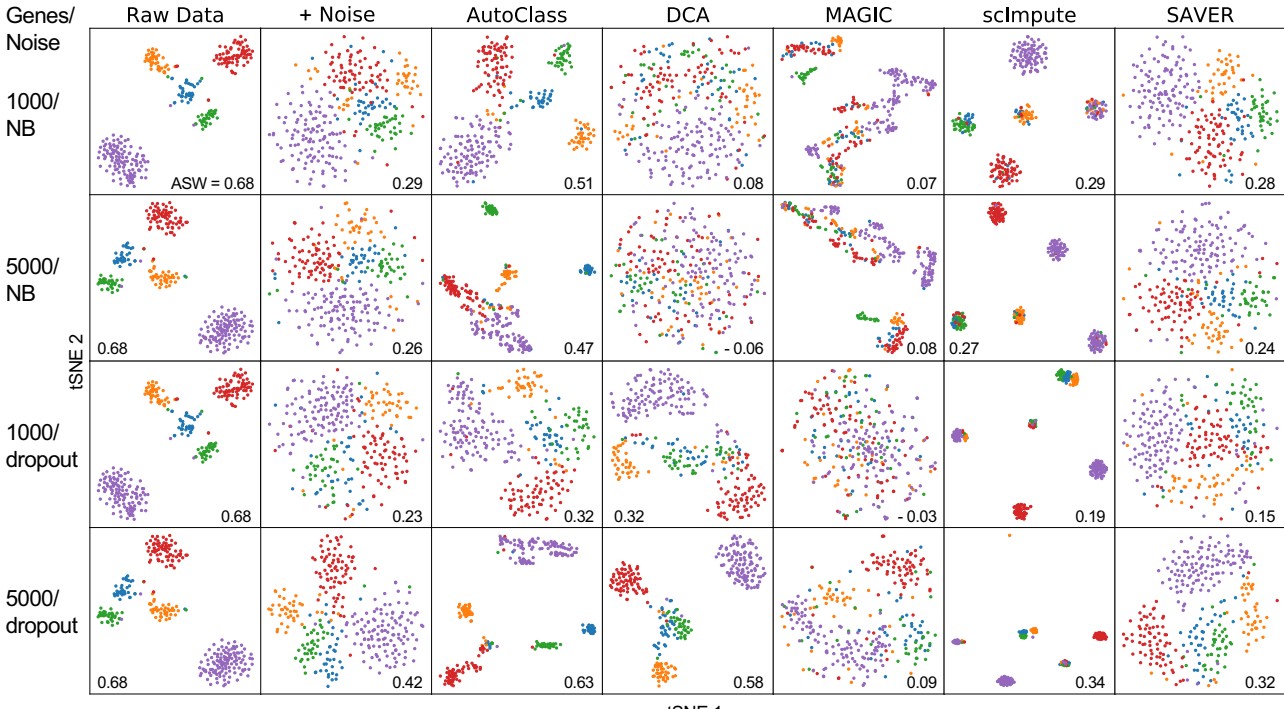

**Fig. 8 Imputation results for Tian et al cell mixture dataset.** The rows are different feature sizes (genes) and additional noise types, i.e. dropout vs NB (Negative Binomial).

AutoClass is stable over different choices of $K$ in pre-clustering as long as $K$ is not extremely far away from the true number of cell clusters. The default value $K = 9$ was used in all datasets in this study except simulated Dataset 8 and 9, since the true number of cell clusters in these two datasets is 2 which is far smaller than default value 9. Hyperparameter $K$ can be chosen based on prior knowledge of the data or statistical methods like elbow method[28] and Silhouette method[14]. $K$ used in Dataset 8 and 9 was the average of estimations by elbow method and Silhouette method.

AutoClass is stable on classifier weight $w$ in the range of 0.1–0.9 (Supplementary Fig. 8). We found that in general classification loss is far smaller than reconstruction loss (Supplementary Fig. 9), to have a better balance between those two losses, we set the default value to be $w = 0.9$. This default value was used in all the datasets in this study.

In addition, overfitting is a common problem in neural network models[29]. Dropout of neurons[30] and sparse connections[29] are common regularization methods. Dropout of neurons in the bottleneck layer is used in AutoClass to prevent overfitting. Interestingly, a relatively high dropout rate in AutoClass also helps to correct batch effect. In the batch effect removal analyses, we set dropout rate to be 0.5 in AutoClass, and to be fair, also in DCA. But DCA was unable to remove batch effect (Fig. 5 and Supplementary Figs. 4, 5). The default dropout rate 0.1 in AutoClass was used in all the other datasets and analyses in this study. Note methods like scVI[31] and Linnorm[32] use parametric distributions to decouple biological signal from batch effect (or other technical variations). AutoClass does so using its special neural nework architecture, i.e., an autoencoder and a classifier (Fig. 1a and "Methods"). Higher dropout rate at bottle neck layer can further reduce overfitting, which changes the relative weight of signal vs noise modeling.

AutoClass hyperparameter settings for all the datasets can be found in Supplementary Tables 1–3.

### Analysis details

*Noise types other than dropout.* Dataset 3–7 and Dataset 9 were generated by manually adding noise to the true data of Dataset 2 and Dataset 8, respectively. The noise was first generated by *Python numpy.random* package with different noise distributions (details in Supplementary Table 3), and then centered (so that noise mean is 0). The noise was then added to true data, all values were rounded to be integers and negative values set to 0, since scRNA-Seq data raw counts are positive integers.

*Highly variable genes.* The highly variable genes in each dataset are ranked by the ratio between gene-wise variance vs mean computed from non-zero values.

*t-Distributed stochastic neighbor embedding (t-SNE).* We applied t-SNE[33] to visualize datasets. We first reduce the number of data dimensions by using the top 50 principle components, and then use *TSNE* function in the *sklearn.manifold* package with default settings to further reduce the dimension to 2 for visualization.

*Batch effect removal score.* Four clustering metrics ASW, ARI, NMI, and PS were used to measure the performance of batch effect correction. We applied ASW to the t-SNE transformed data, and batch effect removal was scored by both cell-type-wise ASW vs 1 – batch-wise ASW (Fig. 5b and Supplementary Fig. 5). Higher values in both dimensions together denote better batch effect removal. ARI, NMI, and PS metrics are used and plotted in the same fashion as ASW. To compute ARI, NMI, and PS, K-means clustering was performed first to obtain cluster labels, which were then compared to batch labels and cell type labels. The batch indices were computed for each individual cell type first, and take weighted sum across cell types. The weight for each cell type is proportional to the number of cells.

**Control methods.** DCA[9] (version 0.2) was downloaded from https://github.com/theislab/dca

MAGIC[15] (version 0.1.0) was downloaded from https://github.com/KrishnaswamyLab/MAGIC

scImpute[11] (version 0.0.5) was downloaded from https://github.com/Vivianstats/scImpute.

SAVER[10] (version 0.3.0) was downloaded from https://github.com/mohuangx/SAVER.

**Real scRNA-seq datasets.** We collected and analyzed multiple real scRNA-Seq datasets from published studies. These datasets have been well established, widely used, and tested as shown in literature. While major technical attributes are summarized in Supplementary Table 1, below are more details.

*Baron study[16].* Human pancreatic islets cells data were obtained from three healthy individuals, which provided gene expression profiles for 17,434 genes in 7729 cells. We filtered out genes expressed in less than 5 cells, removed cell types less than 1% of the cell population. Analysis was restricted to top 1000 highly variable genes. Final dataset contained 7162 cells with eight different cell types.

The raw counts data are available at https://shenorrlab.github.io/bseqsc/vignettes/pages/data.html.

*Villani study[26].* The human blood dendritic data contained 26,593 genes in 1140 cells. We kept batch 1 (plate id: P10, P7, P8 and P9) batch 2 (plate id P3, P4, P13, and P14) cells, and filtered out genes expressed in less than 5 cells. Analysis was restricted to top 1000 highly variable genes. Final dataset contained 768 cells with 4 different cell types in 2 batches. The raw data are available at GEO accession GSE80171 https://www.ncbi.nlm.nih.gov/geo/query/acc.cgi?acc=GSE80171.

*Lake study[20].* Human brain frontal cortex data contained 34,305 genes in 10,319 cells. We filtered out genes expressed in less than 5 cells, removed cell types h less than 3% of the cell population. Analysis was restricted to top 1000 highly variable

genes. Final dataset contained 8592 cells with 11 different cell types. The raw data are available at GEO accession GSE97930 https://www.ncbi.nlm.nih.gov/geo/query/acc.cgi?acc=GSE97930.

*Zeisel study*[21]. Mouse cortex and hippocampus data contained 19,972 genes in 3005 cells. We filtered out genes expressed in less than 5 cells. Analysis was restricted to top 1000 highly variable genes. Final dataset contained 3005 cells with nine different cell types. Annotated data are available at http://linnarssonlab.org/cortex.

*Buettner study*[2]. Mouse embryonic stem cells contained 8989 genes in 182 cells. We filtered out genes expressed in less than 5 cells. Final dataset contained 8985 genes and 182 cells in 3 cells lines. The full dataset was deposited at ArrayExpress: E-MTAB-2805. The normalized data can be obtained from https://www.nature.com/articles/nbt.3102.

*Usoskin study*[19]. Neuronal data contained 17,772 genes in 622 cells. We filtered out genes expressed in less than 5 cells. Analysis was restricted to top 1000 highly variable genes. Final dataset contains 622 cells with 4 different cell types. The normalized data can be obtained from https://www.nature.com/articles/nbt.3102.

*Tian study*[27]. We worked on the scRNA-Seq data on the mixture of five cell lines, containing 13,426 genes in 305 cells. We kept top 1000 and 5000 variable genes for analysis. To evaluate the denoising performance, two types of noise were added to the raw data: (1) negative binomial noise (parameter settings in Supplementary Table 3); (2) dropout noise following a logistic function defined by shape and mid parameter as used in Splatter (parameter settings in Supplementary Table 2). The original dataset was downloaded from https://github.com/LuyiTian/sc_mixology/tree/master/data.

**Simulated scRNA-seq datasets**. Splatter R (version v1.2.2) package was used to simulate scRNA-seq datasets with dropout values. Gaussian noise was manually added when needed. Genes expressed in less than 3 cells were filtered out before analysis. The parameter settings for simulation are summarized in Supplementary Tables 2 and 3.

Dataset 10 was generated using the alternative method used by scImpute study[11]. In Dataset 10, there were 7 cell types and 10,000 cells (or ~1428 cells each), and 10,000 genes in total. For individual genes, the mean expression was randomly drawn from a normal distribution (mean = 1.8, sd = 0.5), the standard deviations drawn from another normal distribution (mean = 0.6, sd = 0.1). Then 50 random genes were set as markers for each cell type, where their mean expression was scaled by a factor from uniform distribution in 1.5–2. Finally, the dropouts or zeros were introduced by the rate following a double exponential function $\exp(-0.1 \times$ expression value).

**Reporting summary**. Further information on research design is available in the Nature Research Reporting Summary linked to this article.

## Data availability

All simulated datasets can be generated using the parameters specified in the "Simulated scRNA-Seq datasets" subsection, all the real datasets are publicly available with URLs and references listed in the "Real scRNA-Seq datasets" subsection above. In addition, simulated and real datasets were provided in the GitHub repository https://github.com/datapplab/AutoClass as demo datasets, ready for analysis.

## Code availability

AutoClass python module, documentation, tutorial with example, and code to reproduce the main results in the manuscript are available online: https://github.com/datapplab/AutoClass[34].

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

## Acknowledgements

The authors are supported in part by the U.S. National Science Foundation grant ABI-1565030 to W.L. We also acknowledge the support and assistant by the Department of Bioinformatics and Genomics at UNC Charlotte in the publication of this work.

## Author contributions

H.L. and W.L. designed the research. H.L. and W.L. developed the method and conducted the research. W.L. supervised the project. H.L. and W.L. prepared the manuscript and C.B. contributed to the writing.

## Competing interests

The authors declare no competing interests.
