## [Peer Review File · Nature Communications]

Reviewers' Comments:

Reviewer #1:

Remarks to the Author:

Existing denoising and imputation methods have strong distribution assumptions which greatly limit their performance and applications. The authors developed a neural network-based method to maximize both noise removal and signal retention. The method is free of distribution assumptions and can model non-linear relationships. They tested their model extensively and the model shows superior performance when compared with state-of-the-art methods in multiple types of scRNA-seq data analysis. The method looks promising and can be of interest to general readers. I have the following comments:

Major:

1. The testing results are mostly based on simulated data. It is a necessary step for model evaluation but not enough. The authors should also test on real datasets and demonstrate the model's utility in discovering real biological insights. One dataset they can use is Tian et al (PMID: 31133762).
2. They should compare their model with Linnorm (PMID: 28981748), a software with similar functions such as data denoising, transformation and normalization, DEG detection, batch effect removal, imputation, highly variable gene detection and clustering, but with distribution assumption.
3. As there is no ground truth in single cell RNA-seq data clustering, it may make sense to compare the concordance (or purity) of the clustering results of all methods.
4. Autoclass is essentially a Blackbox model. It is not clear to me why it does better in all the functions. As the authors mentioned, it has a high dimension of parameters. With limited no. of genes in the scRNA-seq data. The authors must demonstrate the performance is not due to overtraining, and that the results are robust among different size of scRNA-seq data.

Reviewer #2:

Remarks to the Author:

In this paper, the authors propose a supervised autoencoder model called AutoClass for modeling single-cell RNA-seq data. The model uses an encoder network that takes the high dimensional gene expression (20k) and outputs a low-dimensional hidden feature (e.g., $K=9$). Then the hidden features are used to (1) reconstruct the gene expression; (2) predict cell types. Upon training the model, the authors consider the reconstructed gene expression from the model as the imputed or denoised version of the gene expression. The authors claim two main contributions: (a) distribution-free model assumption; (b) the use of cell type labels to help get better cell type clustering. Overall, the paper is well organized and clearly written. However, I have three major concerns on the overall novelty of the method.

1. The authors seem to be confusing the frequentist non-probabilistic objective with the Bayesian distribution-free model assumption. The loss function in their model is mean squared error (MSE), which is equivalent to assuming a univariate and univariate Gaussian likelihood for expression of each gene from the scRNA-seq data. A principled likelihood-free inference involves approximate Bayesian inference over an unknown likelihood function (e.g., Thomas et al., arXiv 2016). Even using MSE loss on modeling scRNA-seq data is not new because SAUCIE by Amodio et al (Nature Methods 2019) also uses the MSE loss as part of their loss function.

2. The authors used the ground-truth cell labels during the model training as part of the classification task but then evaluated the clustering quality in terms of recovering the true cell types. This is circular. Existing scRNA-seq models are unsupervised (e.g., scVI, scVAE, scAlign, Seurat) without knowing the cell type information. Then cluster generated by each model is used to compared with the ground truth cell types. Providing the model with cell type labels as a supervised learning task defeats the purpose of evaluating the clustering qualities such as ARI, which are unsupervised metrics. With supervised model, one can just evaluate the prediction accuracy on the held-out test cells (e.g., scAlign++). Therefore, it is not surprising that AutoClass outperforms other (unsupervised) methods in clustering cells into the ground truth cell types.

That being said, authors did mention that when cell type labels are unknown they use “virtual cell class” from k-means clustering instead but it is unclear in what experiment they used the “virtual cell class” in what evaluation they used the actual ground-truth cell type labels to train the model.

3. As authors mentioned, Batch effect correction is an important topic in scRNA-seq data analysis. However, there is no additional modeling contributions (e.g., scVI) to correct batch effects other than adding dropout of neurons in the neural network hidden layers (which has become a de facto training trick for large neural network anyway). I find that it is unsatisfactory to merely associate the good batch mixing effects with the dropout trick.

I also have two main concerns on the experiments conducted in this study:

1. The authors evaluated their methods using the Splatter-simulated dataset with only 1000 genes and 500 cells. This is unrealistically small scRNA-seq data where the state-of-the-art scRNA-seq can generate at order of 100,000 or even over a million of cells (Svensson et al., 2020). Also filtering out 1000 most variable genes is not an ideal approach since SOTA scRNA-seq deep models such as scVI and scVI-LD can work with the full 20k gene dimension.

2. Authors have shown the benefits of using reconstructed scRNA-seq in differential analysis in terms of t-statistics in identifying the marker genes in Figure 3. Based on the flat ROC curves in panel C there are only a few positive genes, and the vast majority of the genes are in fact negative. It also seems that at low false discover rate (i.e., 1-specificity) around 10% their method is outperformed by DCA and scImpute. Also, Panel f, shows that Macgic has slightly higher median t-statistic than AutoClass and tighter box. Therefore, I don't quite find this analysis help much in promoting AutoClass in the context of existing methods.

References:

- Amodio, M., Dijk, D., Srinivasan, K., Chen, W. S., Mohsen, H., Moon, K. R., et al. (2019). Exploring single-cell data with deep multitasking neural networks. *Nature Methods*, 16(11), 1–17. <http://doi.org/10.1038/s41592-019-0576-7>
- Thomas, O., Dutta, R., Corander, J., Kaski, S., & Gutmann, M. U. (2016, November 30). Likelihood-free inference by ratio estimation. *arXiv.org*.
- Lopez, R., Regier, J., Cole, M. B., Jordan, M. I., & Yosef, N. (2018). Deep generative modeling for single-cell transcriptomics. *Nature Methods*, 15(12), 1–11. <http://doi.org/10.1038/s41592-018-0229-2>
- Svensson, V., Gayoso, A., Yosef, N., & Pachter, L. (2020). Interpretable factor models of single-cell RNA-seq via variational autoencoders. *Bioinformatics (Oxford, England)*, 36(11), 3418–3421. <http://doi.org/10.1093/bioinformatics/btaa169>

Reviewer #3:

None

Reviewer #4:

None

We appreciate all reviewers' work and comments, and we take them seriously. We carried out multiple additional experiments and analyses, addressed all comments and updated the manuscript. With these updates, we feel that this work and the manuscript have been improved substantially.

The revision took us months because of the amount of additional work and also because of major changes in our lives during this period. The first author, Dr. Li, moved back to China with her family and started a new job there. The corresponding author, Dr. Luo, transitioned to a new position at Novant Health.

Reviewer #1 comments and Authors' Responses

Existing denoising and imputation methods have strong distribution assumptions which greatly limit their performance and applications. The authors developed a neural network-based method to maximize both noise removal and signal retention. The method is free of distribution assumptions and can model non-linear relationships. They tested their model extensively and the model shows superior performance when compared with state-of-the-art methods in multiple types of scRNA-seq data analysis. The method looks promising and can be of interest to general readers. I have the following comments:

Thanks for reviewer 1's positive comments and appreciation of the quality and significance of our work. Her/his critical comments were constructive and helped us further improve our work. Please see our detailed responses to the comments below.

Major:

1. The testing results are mostly based on simulated data. It is a necessary step for model evaluation but not enough. The authors should also test on real datasets and demonstrate the model's utility in discovering real biological insights. One dataset they can use is Tian et al (PMID: 31133762).

Done. We added an additional analysis of Tian et al cell mixture dataset (Fig. 8). In addition, we also added another figure to show that AutoClass can discover relevant biological insights (novel marker genes) from real data analyses (Supple. Fig. 3).

We agree that it is necessary to evaluate models/methods on real data. We now include 7 real datasets (Supple. Table 1) in addition to the multiple simulated datasets as to test the method in a wide range of sample/feature sizes, noise types and data quality conditions.

2. They should compare their model with Linnorm (PMID: 28981748), a software with similar functions such as data denoising, transformation and normalization, DEG detection, batch effect removal, imputation, highly variable gene detection and clustering, but with distribution assumption.

We see the reviewer's point here. Linnorm is indeed an excellent method for scRNA-Seq data analysis. However, Linnorm is primarily a normalization and transformation method, and it was only compared to other normalization methods but not denoising methods in the original paper (Yip S. *et al*, 2017). In this work, we present AutoClass as a data denoising and imputation method. Data normalization is important but not the focus of AutoClass. We have not seen Linnorm used as control in other denoising method papers (DCA, SAVER, MAGIC, scImpute) either. Note that both data normalization and denoising can improve downstream analyses including DEG detection and clustering, but these analyses are not part of normalization or denoising methods. Therefore, we feel that Linnorm is less comparable than the control methods we used. Nonetheless, we cited Linnorm in the most relevant context, i.e. the batch effect removal part of Methods section (Page 14 line 22).

In the future, we would be happy to explore more of the use of AutoClass in data normalization, and its potential integration and comparison with Linnorm.

3. As there is no ground truth in single cell RNA-seq data clustering, it may make sense to compare the concordance (or purity) of the clustering results of all methods.

Done. Purity score (PS) is actually one of the metrics we used in comparing the clustering results of all methods (Table 1, Figure 5, Supple. Fig. 4). It was first defined in page 6 line 20. We used ASW too when there is no ground truth on cell clusters.

4. Autoclass is essentially a Blackbox model. It is not clear to me why it does better in all the functions. As the authors mentioned, it has a high dimension of parameters. With limited no. of genes in the scRNA-seq data. The authors must demonstrate the performance is not due to overtraining, and that the results are robust among different size of scRNA-seq data.

Done. The reviewer made a great point here. We tested AutoClass on both real and simulated datasets extensively, which cover a wide range of sample sizes (cell number) and feature sizes (gene number), cell groups and conditions (Supple. Table 1-3). In this revision, we added a new subsection "Large feature size and sample size" in Results (Page 10 line 11) and intentionally added datasets with larger sample size, feature size or both in this revision, including multiple analyses of Tian *et al* dataset, Dataset 10 and 11 (Fig 8 and Supple Fig 11) as to further increase the coverage of different data sizes and conditions.

We understand the concern on potential overtraining problem. It is not the case here because all the results we show are testing results not training results. To further clarify, we updated the Introduction section and described AutoClass as unsupervised or self-supervised method (Page 3 line 5). The ground truth (true data values) are either not known (real data) or not used (simulated data) by AutoClass in denoising or imputation. In addition, true cell type labels, even when available, were not used in AutoClass in all analyses here. Instead it used virtual class labels generated by pre-clustering.

Reviewer #2 comments and Authors' Responses

In this paper, the authors propose a supervised autoencoder model called AutoClass for modeling single-cell RNA-seq data. The model uses an encoder network that takes the high dimensional gene expression (20k) and outputs a low-dimensional hidden feature (e.g., $K=9$). Then the hidden features are used to (1) reconstruct the gene expression; (2) predict cell types. Upon training the model, the authors consider the reconstructed gene expression from the model as the imputed or denoised version of the gene expression. The authors claim two main contributions: (a) distribution-free model assumption; (b) the use of cell type labels to help get better cell type clustering. Overall, the paper is well organized and clearly written. However, I have three major concerns on the overall novelty of the method.

Thanks for reviewer 2's insightful comments and appreciation all of them, no matter positive or negative. These comments help us further improved our work. Please see our detailed responses to the comments below.

1. The authors seem to be confusing the frequentist non-probabilistic objective with the Bayesian distribution-free model assumption. The loss function in their model is mean squared error (MSE), which is equivalent to assuming a univariate and univariate Gaussian likelihood for expression of each gene from the scRNA-seq data. A principled likelihood-free inference involves approximate Bayesian inference over an unknown likelihood function (e.g., Thomas et al., arXiv 2016). Even using MSE loss on modeling scRNA-seq data is not new because SAUCIE by Amodio et al (Nature Methods 2019) also uses the MSE loss as part of their loss function.

Done. The reviewer made a great point here. We agree the use of MSE as part of the loss function implies Gaussian distribution of noise. This is not what AutoClass actually is, and sorry for the confusion. To clarify this important issue, we added multiple analyses (Supple. Fig. 10) and a new subsection (starting from Page 9 line 12) to show that AutoClass works equally well with other forms of reconstruction error, MSE happens to be one of them. This issue was further covered in Discussion section (Page 12 line 3). In addition, we also updated AutoClass method description (Page 12 line 24 and Page 13 Equation 4-5). Note that we have no intention to claim MSE or the loss function as the novelty or contribution of this work.

These new results confirm that AutoClass does not depend on the assumption of Gaussian distribution. Because it works equally well with MAE and other forms of reconstruction error in the form of $|Y_{\text{pred}} - Y_{\text{true}}|^p$ ($p=1, 2, 3$ etc). Note reconstruction error is MAE when $p=1$, and MSE when $p=2$. Another important proof of distribution free property is that AutoClass works well with a wide range of different noise distributions (Fig. 2 and

Supple. Fig. 1) even when MSE was used. Otherwise, it would have only worked well with Gaussian noise, but not other types of noise.

2. The authors used the ground-truth cell labels during the model training as part of the classification task but then evaluated the clustering quality in terms of recovering the true cell types. This is circular. Existing scRNA-seq models are unsupervised (e.g., scVI, scVAE, scAlign, Seurat) without knowing the cell type information. Then cluster generated by each model is used to compared with the ground truth cell types. Providing the model with cell type labels as a supervised learning task defeats the purpose of evaluating the clustering qualities such as ARI, which are unsupervised metrics. With supervised model, one can just evaluate the prediction accuracy on the held-out test cells (e.g., scAlign++). Therefore, it is not surprising that AutoClass outperforms other (unsupervised) methods in clustering cells into the ground truth cell types.

That being said, authors did mention that when cell type labels are unknown they use “virtual cell class” from k-means clustering instead but it is unclear in what experiment they used the “virtual cell class” in what evaluation they used the actual ground-truth cell type labels to train the model.

Done. The reviewer has a great point on the circular issue. Using known cell labels during the model training is not good for downstream clustering analysis. We were aware of this issue. Pre-clustering is the default and used in all AutoClass analyses in this work. All known cell type labels were used for method evaluation and comparison only.

Use of pre-clustering was explicitly described in Methods section, e.g. Page 13 Equation (4) and line 3 (\hat{C}_k is the pre-clustering cell type labels for k clusters), as well as Page 13 line 10-14.

For further clarity, we updated our expression in Page 3 line 2-3 and Page 13 line 11-14.

3. As authors mentioned, Batch effect correction is an important topic in scRNA-seq data analysis. However, there is no additional modeling contributions (e.g., scVI) to correct batch effects other than adding dropout of neurons in the neural network hidden layers (which has become a de facto training trick for large neural network anyway). I find that it is unsatisfactory to merely associate the good batch mixing effects with the dropout trick.

We see reviewer’s point that “there is no additional modeling contributions to correct batch effects”. We actually made no claim on “modeling contribution” in this aspect. We present AutoClass as a method to remove different types of noise and technical artifacts (including batch effect), which is supported by the analyses in this work. As the reviewer said, dropout of neurons is regularly used in the neural network models. We did find relatively high dropout rate (0.5 vs default 0.1) in the bottleneck layer of AutoClass improves batch effect removal (Page 14 line 17-22), but we did not claim that as a novelty or original contribution. Therefore this technical observation is described in our Methods section not in the Abstract, Results or Conclusion sections.

To further clarify this issue, we cited scVI and update our description on batch effect

removal starting at Page 14 line 22. There is no doubt that scVI is an excellent method for normalization and analysis of scRNA-seq data. In the future, we would be happy to explore the potential integration and comparison of AutoClass with scVI, esp. in data normalization.

I also have two main concerns on the experiments conducted in this study:

1. The authors evaluated their methods using the Splatter-simulated dataset with only 1000 genes and 500 cells. This is unrealistically small scRNA-seq data where the state-of-the-art scRNA-seq can generate at order of 100,000 or even over a million of cells (Svensson et al., 2020). Also filtering out 1000 most variable genes is not an ideal approach since SOTA scRNA-seq deep models such as scVI and scVI-LD can work with the full 20k gene dimension.

Done. To address the reviewer's concern on relative small sample size or/and features size, we added a new subsection in Results section, "Large feature size and sample size". With multiple extra datasets (both real and simulated), we showed that (1) the performance of AutoClass was similar or even better when more genes were included or feature size became larger (5000 vs 1000 genes, Fig. 8); (2) AutoClass applies well to datasets with both larger features size (10000 genes) and sample size (10000 cells) (Supple. Fig. 11). In addition, we also added another new Scalability subsection in Results, which shows that AutoClass was highly efficient and scalable.

We also further clarify why we used relative small datasets in many benchmark experiments (Page 10 line 12-18): (1) many genes that are mostly zeros or hardly changes across cells are non-informative for downstream analyses or denoising; (2) small datasets are needed as to complete benchmark experiments on all methods in time (including the slow ones shown in Fig. 7). For the same reasons, the scVI papers focused on the top 558-720 genes in multiple analyses (Lopez, R. et al, 2018) even though it can process datasets of many more genes.

2. Authors have shown the benefits of using reconstructed scRNA-seq in differential analysis in terms of t-statistics in identifying the marker genes in Figure 3. Based on the flat ROC curves in panel C there are only a few positive genes, and the vast majority of the genes are in fact negative. It also seems that at low false discover rate (i.e., 1-specificity) around 10% their method is outperformed by DCA and scImpute. Also, Panel f, shows that Macgic has slightly higher median t-statistic than AutoClass and tighter box. Therefore, I don't quite find this analysis help much in promoting AutoClass in the context of existing methods.

Done. We understand the reviewer's concern here and have updated the DE section with new data. For Fig. 3 a-d, Dataset 8 has 161 true DE genes out of 1000 genes in total (Page 5 line 8), rather than "a few positive genes". Like in real datasets, most genes are non DE genes, but 161 true DE genes would be sufficient for us to evaluate the DE

analysis results. We also carried further analyses on the ROC curves in Fig 3c. At specificity=0.90 or 1-specificity=0.10 (i.e. false positive rate 0.1) as marked by the newly added dashed vertical line in Fig 3c, the ROC curves marked different levels of sensitivity, i.e. 0.72 (True data), 0.61 (AutoClass), 0.52 (DCA), 0.51(scImpute), 0.41 (MAGIC), 0.35 (SAVER)), and 0.30 (Raw data. These results confirmed that AutoClass is the best method in this DE analysis, and achieved the closest performance to the True data. Therefore, we update “Differential expression analysis” subsection correspondingly. We appreciate the reviewer’s comments here, which helped to make our results more convincing.

We agree that DCA and scImpute achieved higher sensitivity (than AutoClass) at even higher specificity threshold (1- specificity <0.05). But we always need to balance specificity and sensitivity, and that’s why ROC curves and AUC scores are frequently used, rather than specificity or sensitivity alone, in performance evaluation. Indeed, AutoClass is the best among these methods by ROC and AUC (Fig 3c-d, and Page 5 line 12-17).

For marker gene analysis, we agree that Fig. 3f alone is not impressive. We have updated the paragraph (starting at Page 5 line 22) and shown that AutoClass is the only method that improves both fold changes (Fig. 3e) and t-statistics (Fig. 3f). We also added one paragraph (starting at Page 6 line 3) as to show how AutoClass improves the data quality and DE analyses by analyzing multiple specific examples of potential marker genes (Supple. Fig. 3).

Reviewers' Comments:

Reviewer #1:

Remarks to the Author:

The authors have addressed my comments very well. I recommend publication of the manuscript.

Reviewer #2:

Remarks to the Author:

I like to thank the authors for addressing my comments with great efforts. However, my main concern about the novelty of the method and new contributions remains. Please see my detailed comments in responding to authors' response below.

1. For my comments on the "distribution-free" claim of the paper, authors responded with experiments trying out different likelihoods (e.g., Gaussian, Poisson, NB, etc) and different norms (e.g., $|Y_{\text{pred}} - Y_{\text{true}}|^p$ for $p = 1, 2, 3$). This shows that the autoencoder is robust to different loss functions, but it does not make the model "distribution-free" and did not actually address my first comment. In my first comment, I referred the authors to a *real* likelihood-free work done by Thomas et al 2020 (see ref below). To claim "distribution-free", you cannot assume any form of known density functions; rather one needs to work with a non-differentiable "black-box" likelihood function with no closed-form expression (e.g., a neural network likelihood). Deep learning Bayesian researchers have come up ways to approximate this black-box distribution by sampling. Therefore, authors should really drop all the claims in the text on "free of distribution assumption", "does not presume any type of data distribution", etc about their model. It is correct if authors claim that their model (or autoencoder in general) is robust to different distribution assumptions or reconstruction loss functions.

2. Now once the above mis-claim about likelihood-free is resolved, my following question is that what is the methodological novelty that AutoClass has *in contrast to SAUCIE* (Amodio et al., NatMet 2019)? SAUCIE has the same autoencoder framework with a bottleneck encoder layer, and it also does denoising and cell type identification using the binary encoding layer. Authors need to clearly enlist the novel contributions in the context of the body of the literatures on scRNA-seq deep learning methods especially SAUCIE.

3. For my comments on DE analysis, I strongly suggest that the authors should switch to precision-recall curves instead of ROC due to highly unbalanced true DE genes (i.e., as authors said only 161 out of 1000 genes are true DE). ROC is known to be ill-calibrated for unbalanced data. In real practice, higher precision ($TP/(TP+FP)$) at a fixed recall rate is more useful for downstream experimentation validations.

Reference:

1. Thomas, O., Dutta, R., Corander, J., Kaski, S. & Gutmann, M. U. Likelihood-Free Inference by Ratio Estimation. Bayesian Anal 1, (2020).
2. Amodio, M. et al. Exploring single-cell data with deep multitasking neural networks. Nature Methods 16, 1-17 (2019).

Reviewer #2 comments and Authors' Responses

I like to thank the authors for addressing my comments with great efforts. However, my main concern about the novelty of the method and new contributions remains. Please see my detailed comments in responding to authors' response below.

Response:

We do thank Reviewer 2 for her/his time and detailed comments. However, we feel that Reviewer 2 made largely misleading and wrong comments this round.

*1. For my comments on the “distribution-free” claim of the paper, authors responded with experiments trying out different likelihoods (e.g., Gaussian, Poisson, NB, etc) and different norms (e.g., $|Y_{pred} - Y_{true}|^p$ for $p = 1,2,3$). This shows that the autoencoder is robust to different loss functions, but it does not make the model “distribution-free” and did not actually address my first comment. In my first comment, I referred the authors to a *real* likelihood-free work done by Thomas et al 2020 (see ref below). To claim “distribution-free”, you cannot assume any form of known density functions; rather one needs to work with a non-differentiable “black-box” likelihood function with no closed-form expression (e.g., a neural network likelihood). Deep learning Bayesian researchers have come up ways to approximate this black-box distribution by sampling. Therefore, authors should really drop all the claims in the text on “free of distribution assumption”, “does not presume any type of data distribution”, etc about their model. It is correct if authors claim that their model (or autoencoder in general) is robust to different distribution assumptions or reconstruction loss functions.*

Response (concise):

Loss function does not equal to likelihood. Reviewer 2 mixed up these concepts. We use loss function but not likelihood to optimize AutoClass model. AutoClass makes no assumption on the data distribution, and it has been fully tested in data generated with various distributions (Fig. 2, Supple. Fig. 1 and Supple. Fig. 10). Note these testing data distributions are not AutoClass's assumption!! Please see the detailed description on this issue below.

Additional clarification:

Loss function is not the same as likelihood. Maximum likelihood estimation (MLE) is just one method to fit models, where (log) likelihood is used as the loss function.

Loss functions are measurements of model misfit, and are more general than MLE. Loss functions don't have to be likelihood or interpreted as likelihood. There are many commonly used loss functions that are not likelihood. For example, the well known SVM uses hinge loss, which is not MLE or likelihood based. Other loss functions including Log-Cosh Loss, Quantile Loss or Huber loss are not likelihood based or does not have obvious MLE interpretation. All of these loss functions have closed-form expression and are differentiable and widely used in machine learning.

Reviewer 2 doesn't seem to understand the difference between these two concepts and misinterprets our loss function as likelihood.

Throughout the manuscript, we use the term “loss function” instead of likelihood or MLE. The loss function measures how well the predictions fit the expected or actual values. It

does not take assumption on any specific distribution, nor depends on any particular form of probability density because we don't use likelihood. Because MSE as the reconstruction error may be taken/interpreted as (but not necessarily is) MLE of Gaussian error, we added multiple analyses (Supple. Fig. 10) as to show that other types of reconstruction loss/error would work without an interpretation of likelihood. Hence even MSE should not be take as MLE here . A similar example, you can calculate mean and standard deviation for data with any distribution forms, not just those with normal distribution.

Note the term “distribution assumption free” is not exactly the same as “distribution free”. All data including training or testing data have some distribution, a method made no assumption on the form of data distribution does not mean it assumes the data has no distribution at all.

*2. Now once the above mis-claim about likelihood-free is resolved, my following question is that what is the methodological novelty that AutoClass has *in contrast to SAUCIE* (Amodio et al., NatMet 2019)? SAUCIE has the same autoencoder framework with a bottleneck encoder layer, and it also does denoising and cell type identification using the binary encoding layer. Authors need to clearly enlist the novel contributions in the context of the body of the literatures on scRNA-seq deep learning methods especially SAUCIE.*

Response (concise):

SAUCIE is very different from AutoClass in both architectures and target applications. SAUCIE aims for multitasking (clustering, batch correction, visualization, and imputation), AutoClass focus on data cleaning/denoising. SAUCIE is a sparse autoencoder, while AutoClass merges a regular autoencoder with a classifier. SAUCIE has not been used as control in other denoising method papers (DCA, SAVER, MAGIC, scImpute).

Additional clarification:

The reviewer insisted that AutoClass is similar to SAUCIE because:

(a) both method use MSE. We've already showed that AutoClass can use a series of reconstruction loss other than MSE (Equation 5 and Supple. Fig. 10). In addition, reconstruction loss is only part of the loss function for AutoClass (Equation 4). Note we never claimed MSE (or loss function) as a novelty/contribution, and it is commonly used in deep learning. The reviewer seems to be biased towards such trivial or less relevant issues (also see her/his comments on batch effect and on gene/cell numbers related to scVI in Round 1).

(b) both papers described multiple types of analyses. Note these analyses are not part of AutoClass or control methods here, but the downstream analyses on the data denoised/cleaned by these methods. In opposite, SAUCIE itself cover multiple types of analysis.

In addition, for both comment 1 and 2: The loss function of AutoClass has two parts: classifier loss (cross-entropy or CE) and the autoencoder loss or reconstruction error (RE). Reviewer 2 seemed to have focused solely on the autoencoder loss part and ignored the classifier loss so far. However, one major difference between AutoClass and other autoencoder based methods is exactly the addition of the classifier part (Figure 1).

3. For my comments on DE analysis, I strongly suggest that the authors should switch to precision-recall curves instead of ROC due to highly unbalanced true DE genes (i.e., as authors said only 161 out of 1000 genes are true DE). ROC is known to be ill-calibrated for unbalanced data. In real practice, higher precision ($TP/(TP+FP)$) at a fixed recall rate is more useful for downstream experimentation validations.

Response:

It is widely known that only a small portion of all genes (a few dozen up to 1-2000 out of ~30000 genes) are differentially expressed (DE) in almost all transcriptome studies. By Reviewer 2's standard, essentially all DE analyses are highly unbalanced in DE vs non-DE genes. However, ROC curve and AUC are routinely used in both single cell and bulk RNA-Seq DE analyses:

<https://academic.oup.com/nar/article/45/22/e179/4160405>

<https://www.sciencedirect.com/science/article/pii/S2405471219300808>

<https://genomebiology.biomedcentral.com/articles/10.1186/s13059-016-0940-1>

<https://bsapubs.onlinelibrary.wiley.com/doi/full/10.3732/ajb.1100340>

Note the second URL above is a paper recently published by Dr. Nir Yosef recently.

Authors' Additional Arguments

We claimed three major contributions/novelties of this work as shown in the Title and Abstract:

- 1) AutoClass effectively cleans a wide range of noises and artifacts, which no other method can do;
- 2) AutoClass outperforms the state-of-art methods in multiple types of analyses;
- 3) AutoClass is robust on key hyperparameter settings.

None of these contributions are affected by Reviewer 2's comments. Even if there was distribution assumption (as s/he claimed), it is indisputable that AutoClass is robust and effectively removes a widely range of noises/artifacts with different distribution forms. Given these results, the argument on distribution assumption or not is really trivial. And we can make minimal updates to make that issue clearer if needed. In addition, Reviewer 1 is happy with the current version and gave the recommendation for publication of this work.

As a side note, Review 2 made multiple other misleading (and obviously biased) comments in the last round too, including comment 3 Batch effect and the comment on the number of genes and cells using scVI and scVI-LD as example. Please see our point-to-point response for details.